# Aberrant autophagosome formation occurs upon small molecule inhibition of ULK1 kinase activity

Maria Zachari⬥, Marianna Longo, Ian G Ganley⬥

**Autophagy is a crucial homeostatic mechanism that mediates the degradation of damaged or excess intracellular components. Such components are engulfed and sequestered into double membrane autophagosomes, which deliver their contents to lysosomes for degradation. Autophagy plays a role in numerous human disorders and its pharmacological targeting by small molecules offers therapeutic potential. The serine/threonine kinase ULK1 (and its homologue ULK2) is the most upstream component of the autophagic machinery and is required for autophagy initiation. Here, we use the most selective and potent published ULK1 inhibitors to gain insights into ULK1 kinase function during autophagy. Treatment with all inhibitors blocked autophagy but also resulted in the limited formation of initial autophagosome-like structures, which appeared abnormal in size and did not traffic to lysosomes. We found that upon ULK1 inhibition, phosphatidylinositol-3-phosphate–binding proteins are still recruited to forming autophagosomes, implying that ULK1 activity is not essential for VPS34 activation. We conclude that the kinase activity of ULK1 is important in regulating autophagosome maturation, by the phosphorylation of currently unidentified key substrates.**

## Introduction

Autophagy is a crucial homeostatic mechanism regulating lysosome-dependent degradation and recycling of intracellular components. The most prevalent type of autophagy is called macroautophagy and is the focus of this study. Macroautophagy (simply termed autophagy from now on) promotes cell survival and cell health, and in general is considered a disease-preventing mechanism (1, 2). In the context of cancer, autophagy has a complicated role, as under normal conditions it exhibits a tumour-suppressive role but upon cancer development multiple cancer cell types rely on autophagy to survive within the tumour microenvironment (3, 4). The process of autophagy involves the de novo generation of a double membrane organelle called an autophagosome, which grows to engulf cytosolic cargo before

closing to fuse with the lysosome (5). Autophagosome biogenesis is a complex multistage process and requires the participation of numerous proteins, which often work in complexes. The multiple stages of autophagy include initiation, phagophore elongation and cargo recognition, autophagosome closure, and finally fusion with lysosomes. A key serine/threonine kinase regulating initiation of autophagosome formation is Unc-51 like autophagy-activating kinase 1 (ULK1) (and its homologue ULK2). ULK1 is in a constitutive complex with Autophagy related (ATG) 13 (ATG13), FAK family interacting protein of 200 kD (FIP200) and ATG101, which together is known as the ULK1 complex (6, 7, 8, 9). The role of ULK1 during autophagy has been extensively studied in the context of amino acid starvation, where it has been shown (in multiple cell lines and tissues) that in a fed state, ULK1 is suppressed by direct Mechanistic target of rapamycin complex 1 (mTORC1)-mediated phosphorylation at multiple sites (including serine 757 in murine or 758 in human ULK1) (10). Upon amino acid starvation, mTORC1 is inhibited, resulting in dephosphorylation of ULK1 and subsequent activation of its catalytic activity. Upon activation, ULK1 is reported to phosphorylate multiple downstream autophagy targets (including itself at threonine 180 and ATG13 at serine 318) to mediate initiation of autophagy (11, 12). Among these, are components of the second autophagy initiation complex, the vacuolar protein sorting (VPS) 34 (VPS34) complex. VPS34 is a class-III phosphatidylinositol (PI)-3 kinase and when in complex with Beclin-1, ATG14L, and VPS15 (called the VPS34 complex) produces an autophagy-specific pool of phosphatidylinositol-3-phosphate (PI3P) that is required for omegasome formation—an ER-connected platform from which autophagosomes arise (13, 14, 15, 16). PI3P production recruits PI3P-binding proteins such as WD-repeat protein interacting with Phosphoinositides (WIPI2) and Zinc Finger FYVE domain-containing protein 1 (DFCP1) (16, 17, 18). The role of these proteins is poorly understood, although one known function of WIPI2 is to bind and recruit ATG16L1—an essential protein for autophagosome elongation and lipidation of the autophagosomal membrane microtubule associated protein 1 light chain 3 (MAP1LC3)—to the autophagosome forming sites (19). The ULK1 complex is upstream of the VPS34 complex and ULK1 kinase activity is believed to be key for VPS34 complex recruitment and activation (13, 15). During autophagosome biogenesis, the early autophagy initiating complexes (ULK1 and VPS34), as well as WIPI2 and DFCP1, are recruited

---

Medical Research Council Protein Phosphorylation and Ubiquitylation Unit, University of Dundee, Dundee, UK

Correspondence: i.ganley@dundee.ac.uk

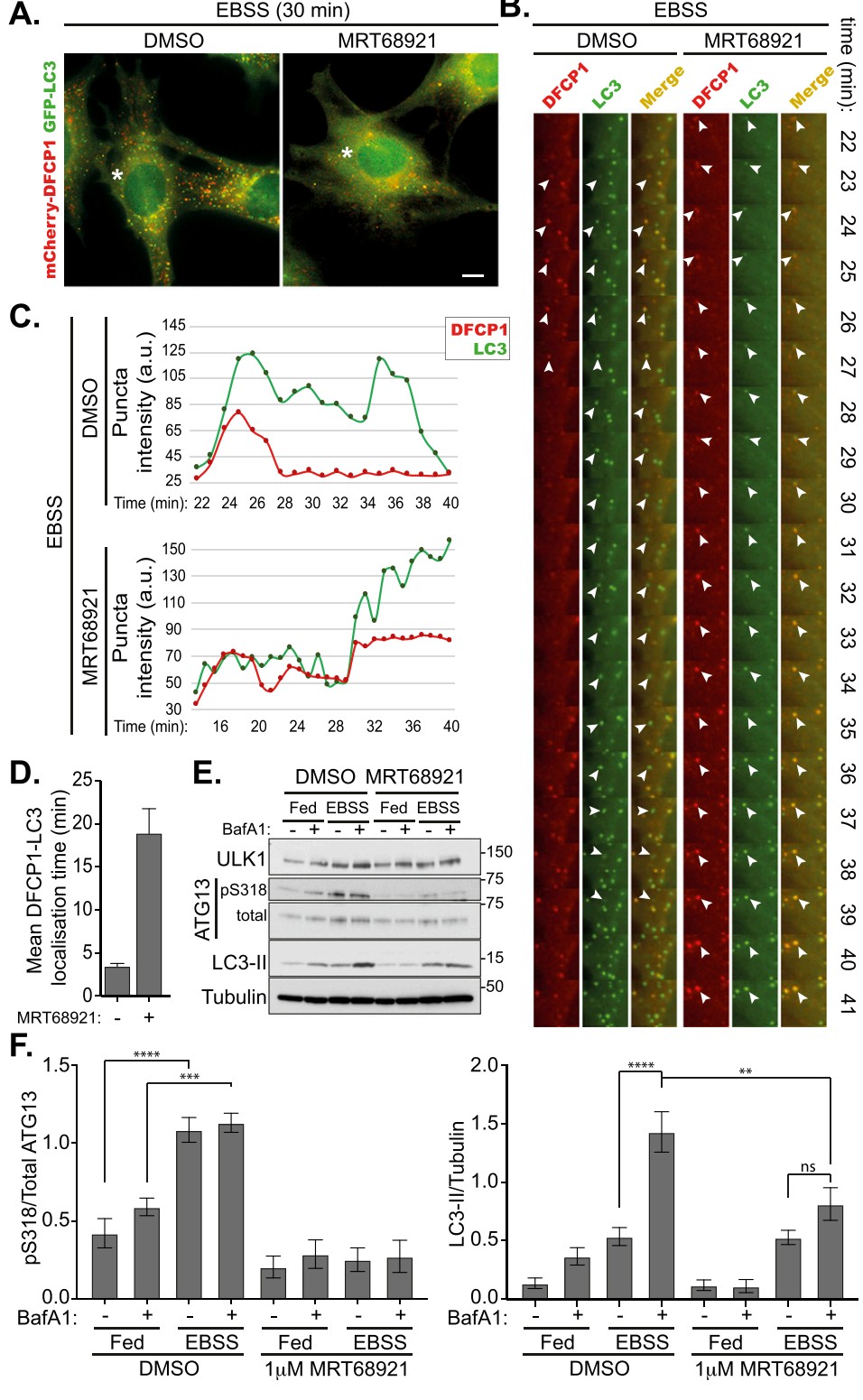

**Figure 1. Stalled autophagosomes upon inhibition of ULK1 with MRT68921 are positive for DFCP1 and LC3.**
**(A)** MEF cells expressing GFP-LC3 and mCherry-DFCP1 were pretreated for 15 min with 1 μM MRT68921 or DMSO followed by treatment with EBSS or EBSS and 1 μM MRT69821. The cells were then immediately subjected to live imaging using a Nikon Eclipse Ti wide-field microscope. **(B)** Representative images of cells are shown at 30 min in and asterisk marks the section highlighted in panel (B). The movies are provided as supplementary information (Videos 1 and 2). Scale bar, 10 μm. **(A, B)** Montage of the marked areas from the movies shown in (A) from minutes 22–41. Arrowheads indicate colocalisation between GFP-LC3 and mCherry-DFCP1. **(B, C)** Graphical representation of the maximum intensity of the mCherry-DFCP1 (red) and GFP-LC3 (green) puncta indicated with arrows in (B). **(D)** Quantitation of the mean time for which GFP-LC3 and mCherry-DFCP1 signals colocalised on the same punctate structure (from n = 4 events, bars represent mean with SEM). **(E)** Immunoblot of WT MEF cells treated with EBSS and 1 μM MRT68921 as indicated. **(F)** Quantitation of (E). pS318-ATG13 levels were normalised to total ATG13 (left) and LC3-II levels were normalised to Tubulin (right). Data represent mean of n = 4 ± SEM. Statistical analysis was performed with two-way ANOVA and a Tukey's multiple comparisons test. * = $P < 0.05$, ** = $P < 0.01$, **** = $P < 0.0001$ and ns, not significant.
Source data are available for this figure.

transiently to nascent autophagosomes and once they have completed their roles, they dissociate from it (20, 21). Given the key role of ULK1 in autophagy initiation and the fact that numerous cancer cells depend on autophagy for survival, inhibition of ULK1 with small-molecule inhibitors makes it a promising target for cancer drug discovery (21, 22, 23). Our laboratory has previously published a study characterising a potent ULK1 inhibitor, MRT68921, which revealed that inhibition of ULK1 in cells blocks autophagic flux (24). Unexpectedly, instead of complete abolishment of

autophagosomes upon MRT68921 treatment, we observed that autophagosomes were still forming but they were abnormally bigger in size and appeared to be stalled (24). These findings suggested that ULK1 kinase activity is regulating later stages of autophagosome formation as well as initiation. Recently two more structurally distinct small-molecule ULK1 inhibitors were published (SBI0206965 and ULK-101) and shown to inhibit ULK1 and autophagy in cells (25, 26). Importantly, these inhibitors display distinct selectivity profiles and in this study, we aimed to gain more insights into the kinase function of ULK1 during autophagy by comparing the common effects of these potent and selective ULK1 inhibitors.

## Results

### Inhibition of ULK1 with MRT68921 results in stalled omegasomes

We have previously shown that ULK1 inhibition with MRT68921 results in stalled autophagosome formation and not abolishment, even though ULK1 is hierarchically the most upstream component of the autophagic machinery (24). To gain a more comprehensive comparison of the forming autophagosomes in normal versus ULK1-inhibited conditions, we performed live imaging using MEF cells stably expressing the early autophagy protein DFCP1 (tagged with mCherry at its N-terminus) and LC3 (tagged with GFP at its N-terminus). For expression levels see Fig S1. DFCP1 marks the transient omegasome, a structure that acts as a cradle for the forming autophagosome (16), whereas LC3 is recruited shortly after DFCP1 and becomes trapped on the inner membrane of the formed autophagosome until its degradation inside the lysosome. To induce autophagy, we used Earle's balanced salt solution (EBSS) (an amino acid starvation medium) and immediately proceeded with live imaging. Initially, the GFP-LC3 signal appeared diffused in the cytoplasm but after 15 min of EBSS treatment GFP-LC3 formed numerous punctate structures, which continued to increase in number in a time-dependent manner (Fig 1A and B and Video 1). mCherry-DFCP1 appeared to already form punctate structures from the beginning, which could be representative of its known localisation on Golgi-derived membranes under basal conditions (16). Nevertheless, a pool of mCherry-DFCP1 was indeed contributing to newly forming autophagosomes, as suggested by its colocalization with GFP-LC3. As expected, upon starvation, mCherry-DFCP1 colocalised with GFP-LC3–positive puncta for a short period of time (~3–5 min, Fig 1B–D), followed by dissociation of GFP-LC3, likely as a fully formed autophagosome, as previously reported (16). Following dissociation, the autophagosomes remained positive for GFP-LC3 and motile in the cytoplasm (Video 1 and Fig 1B–D). To understand the effect of ULK1 inhibition on newly forming autophagosomes, we repeated the experiment in the presence of 1 μM MRT68921. GFP-LC3 puncta formation took much longer to form following ULK1 inhibition but upon 30 min of treatment, some GFP-LC3-positive autophagosomes started to appear, which colocalised with mCherry-DFCP1 (Fig 1A and Video 2). This suggests that autophagosomes initiate in a canonical manner even though ULK1 is inhibited. Regardless, GFP-LC3–positive autophagosomes remained in the omegasome structures for a long period of time (over 20 min, Fig 1B–D), and instead of dissociating they increased in size and displayed reduced mobility

(Videos 1 and 2). Inhibition of ULK1 with MRT68921 was confirmed by Western blot analysis of phospho-ATG13 (at serine 318), a well characterised substrate of ULK1, and a significant block in LC3-II flux (Fig 1E and F). Taken together, these data suggest that upon ULK1 inhibition, autophagosomes are still forming within omegasomes, but they have defects in dissociating from the cradle, mobilisation, and size regulation. Importantly, the presence of DFCP1, which requires PI3P for localisation, suggests that VPS34 is still active despite the fact that ULK1 kinase activity is inhibited and is examined in more detail below.

### Cells expressing kinase-dead ULK1 have phenotypic similarities with MRT68921-mediated ULK1 inhibition

We have previously shown that cells expressing a MRT68921-resistant mutant of ULK1 (M92T) are able to rescue the MRT68921-induced autophagy defect, which strongly suggested the phenotype is specifically due to ULK1 inhibition and not an off-target effect (24). To further validate this phenotype, we expressed a kinase-dead form of ULK1 (K46I, (27)) in double *ULK1/2* knockout MEFs (*ULK1/2* DKO). As shown in Fig 2A, in WT MEFs, endogenous ULK1 and LC3 translocate to numerous small punctate structures, which are partially colocalising depending on the stage of autophagosome formation. Upon, combination of EBSS and MRT68921 treatment, enlarged stalled autophagosomes positive for ULK1 and LC3 were observed, as previously demonstrated (24). In *ULK1/2* DKO MEFs that are treated with EBSS and MRT68921, the appearance of LC3 puncta is vastly diminished, demonstrating that their formation is dependent on ULK1. When we stably expressed WT Flag-ULK1 in the DKO cells, ULK1 and LC3 again translocated to small punctate structures upon EBSS treatment, suggesting that autophagy is rescued by the presence of WT Flag-ULK1 in the *ULK1/2* DKO MEFs (Fig 2A). However, when we expressed K46I Flag-ULK1 in the *ULK1/2* DKO MEFs, although we still observed autophagosomes forming, they were bigger in size and comparable with the ones appearing upon EBSS + MRT68921 treatment in WT ULK1/2 MEFs (Fig 2C). Equal expression of the ULK1 constructs was confirmed by Western blotting and is shown in Fig 2D. The above data further support that the MRT68921-induced phenotype is caused specifically by ULK1 kinase activity inhibition. Moreover, these data show that loss of ULK1 protein does not have the same phenotype as loss of ULK1 kinase activity. Loss of ULK1 (and the redundant ULK2) results in dramatic loss of LC3 punctate structures, supporting the notion that ULK1 is indeed the most upstream autophagy component. In contrast, loss of ULK1 kinase activity (either by genetic modification or pharmacologically) still results in recruitment of the autophagic machinery and appearance of autophagosome-like structures, although these are aberrantly larger in size and display reduced lysosomal flux (Figs 1E and F, 2, and 3 and (24)). This is in support of previously published data showing that catalytically dead ULK1 blocks autophagy (27, 28).

### ULK-101 and SBI0206965 treatments result in a phenotype similar to MRT68921

MRT68921 was the first small-molecule cellular inhibitor of ULK1 reported at the time of its publication. Following this, two other structurally distinct small molecules, namely, SBI0206965 and ULK-101 (See Fig S2), were reported by other groups to potently inhibit

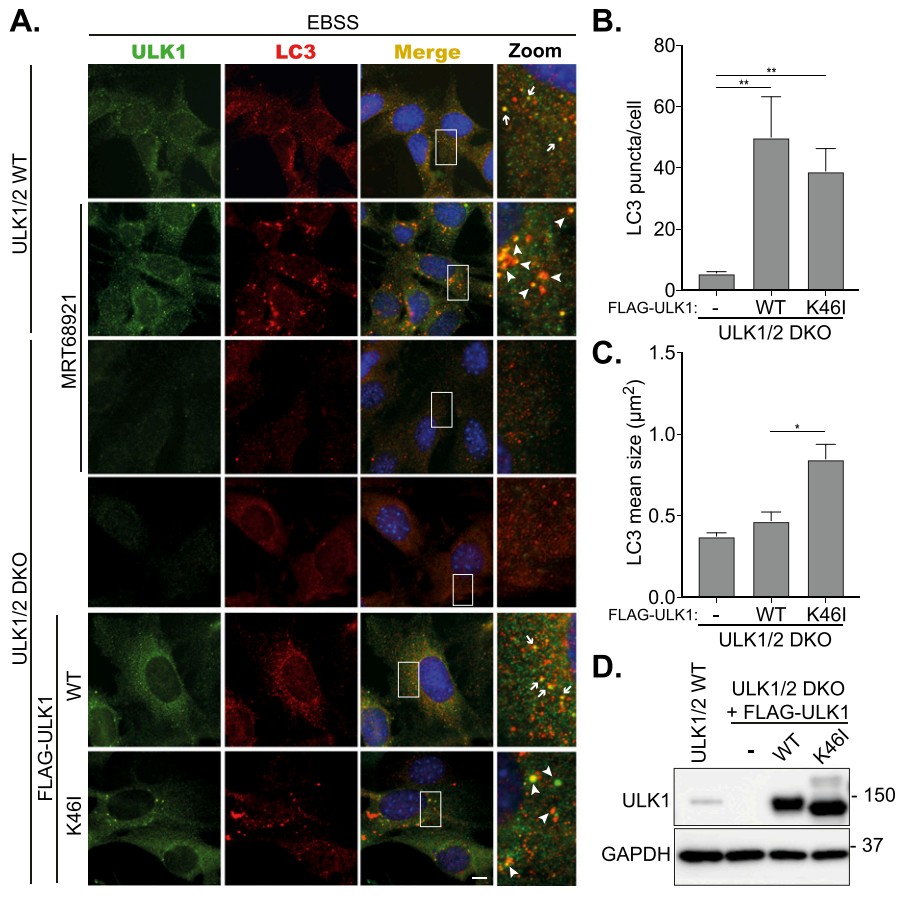

**A.**

**B.** LC3 puncta/cell

**C.** LC3 mean size (μm²)

**D.** ULK1 / GAPDH — 150 / 37

**Figure 2. Expression of kinase-dead ULK1 results in aberrant autophagosomal structures.**
**(A)** WT MEF cells or *ULK1/2* double KO (DKO) expressing Flag-ULK1 WT or Flag-ULK1 K46I (kinase-dead mutant) were treated with EBSS or EBSS and 2 μM MRT68921 for 1 h as indicated. Cells were subsequently fixed and stained with antibodies against ULK1 (green) and LC3 (red). Arrows mark conventional phagophores, whereas arrowheads mark abnormal structures. Scale bar, 10 μm. **(B, C)** Quantitation of LC3 puncta number (B) and size (C) in *ULK1/2* DKO cells and *ULK1/2* DKO cells expressing Flag-ULK1 WT or Flag-ULK1 K46I, treated with EBSS for 60 min. Data represent mean of n = 3 ± SEM. Statistical analysis was performed with one-way ANOVA and a Tukey's multiple comparisons test.* = *P* < 0.05 and ** = *P* < 0.01. LC3 puncta size was calculated manually using the "area" tool in NIS Elements. **(D)** Immunoblot of lysates from cells used in (A) showing ULK1 expression levels.
Source data are available for this figure.

ULK1 and autophagy in cells ([25], [26]). In the study characterising SBI0206965, the authors used the CYTO-ID autophagy detection kit and prolonged treatments to study its effects on autophagic flux. Although the compound blocked autophagic flux, it was unclear whether SBI0206965 treatment would also result in stalled autophagosome formation ([25]). In the study describing ULK-101, the authors performed immunoblot-based LC3 flux assays to show that autophagy is inhibited. They also used immunofluorescence imaging to monitor DFCP1 and ATG12 structures and observed a reduction in their number upon ULK-101 treatment ([26]). Again though, it was unclear from this study if ULK-101 also led to stalled autophagic structures. We thus compared the three inhibitors side-by-side to test whether they displayed phenotypic differences in autophagosome structures. Initially, we performed immunoblotting to identify concentrations and time points where ULK1 activity is sufficiently suppressed and LC3 flux is inhibited, as well as to confirm the activity status of upstream autophagy-regulating kinases (mTORC1 and AMPK) are not affected. As shown in Fig 3A, both SBI0206965 and ULK-101 (at 10 and 1 μM, respectively, for 60 min) significantly inhibited the ULK1-mediated EBSS-induced increase in pS318-ATG13 levels (quantified in Fig 3B). MRT68921 was also included, at 1 μM, where it inhibited ULK1 as expected (Fig 3A and B). Autophagic flux was also impaired by all of the three ULK1 inhibitors, as shown by lipidated LC3 levels (LC3-II) in the presence or absence of BafA1 (Fig 3A and C) and by flow cytometry of cells

expressing the tandem mCherry-GFP-LC3 reporter (([29]) and Fig 3C). Out of the three inhibitors, MRT68921 appeared to be the most potent in inhibiting LC3 flux by Western blot, but flux was also significantly reduced with both SBI0206965 and ULK-101 (Fig 3A–C). It is worth mentioning here that mTORC1 and AMPK activities (as monitored by pS757 ULK1 and pS555 levels, respectively) were not significantly affected upon treatment with MRT68921 and ULK-101, at least at the concentrations and time points tested (Figs 3A and S3A and B). A nonsignificant reduction in pS555 ULK1 levels was observed upon SBI0206965 treatment in Fed conditions, implying potential AMPK inhibition (Figs 3A and S3B), which has also been reported elsewhere ([30]). However, given that AMPK phosphorylation of ULK1 (pS555) is dramatically reduced under these autophagy-inducing conditions, we assume that any AMPK inhibitory effects of SBI0206965 will have a negligible impact on ULK1 in this instance. As with the other inhibitors, mTORC1 activity was unaffected by SBI0206965 treatment (Figs 3A and S3A). In addition, we found that treatment of cells with the three inhibitors did not appreciably disrupt the ULK1 complex itself, as determined by co-IP of ATG13 and FIP200 with ULK1 (Fig S3C). Because we confirmed that ULK1 inhibition was achieved with ULK-101 and SBI0206965 in our hands, we moved on to compare their effects on autophagosome formation. For this purpose, cells were treated with DMSO or the ULK1 inhibitors in Fed or EBSS starvation conditions and then co-stained for ULK1 and LC3, to monitor both early and late

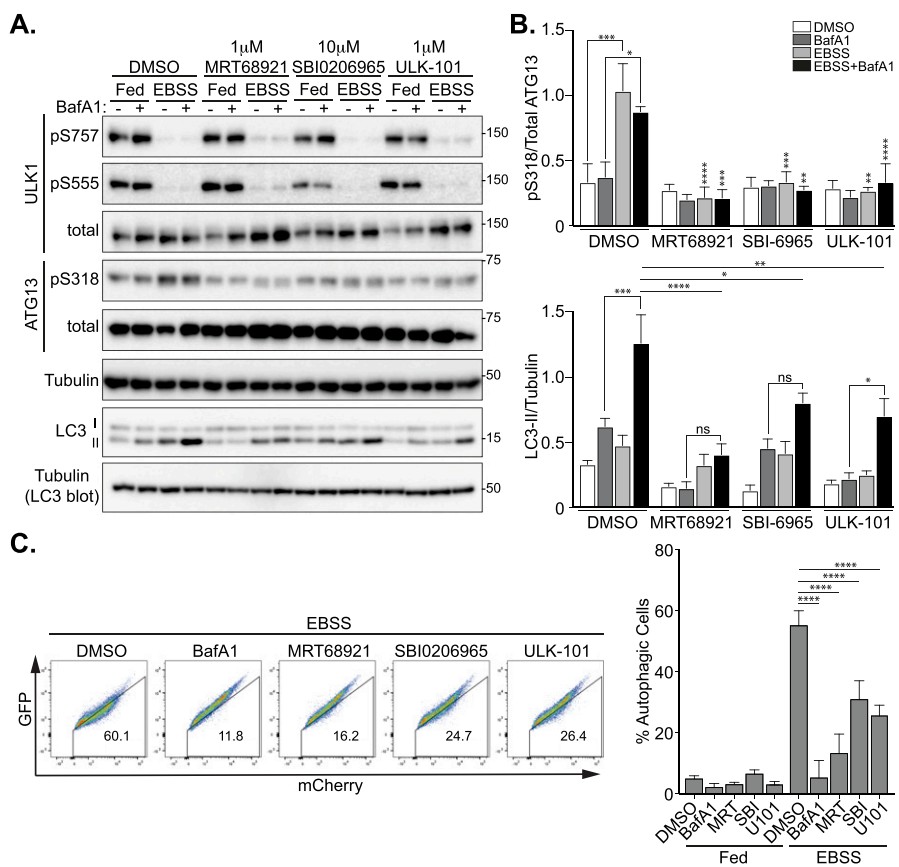

**Figure 3. Distinct ULK1 inhibitors impair autophagy.**
**(A)** Immunoblot of lysates from MEF cells treated with EBSS, 50 nM BafA1, 1 μM MRT68921, 10 μM SBI0206965, and 1 μM ULK-101 for 1 h as indicated. **(B)** Quantitation of pS318-ATG13 levels normalised to total ATG13 (top) and LC3-II levels normalised to α-Tubulin (bottom). Data represent mean of n = 4 ± SEM. Statistical analysis was performed with two-way ANOVA and a Tukey's multiple comparisons test. * = *P* < 0.05, ** = *P* < 0.01, *** = *P* < 0.001, **** = *P* < 0.0001 and ns, nonsignificant (unless indicated comparisons of each treatment are with the relevant DMSO control). **(C)** Representative flow cytometry assay of MEF cells expressing mCherry-GFP-LC3 treated with either DMSO or 50 nM BafA1, or 4 μM MRT68921, or 20 μM SBI0206965, or 2 μM ULK-101 in EBSS for 4 h as indicated, where a decrease/increase in GFP and mCherry expression can be observed and quantified to measure the percent of cells undergoing autophagy. The number in each panel indicates % of cells undergoing autophagy. Quantitation of the flow data is shown on the right and represents mean of n = 3 ± SEM. Statistical analysis was performed with one-way ANOVA and a Tukey's multiple comparisons test. **** = *P* < 0.0001.
Source data are available for this figure.

autophagosomal structures. As expected, in the absence of an autophagy stimulus, the ULK1 inhibitors had little effect on ULK1 and LC3 puncta formation (Fig S4). However, after amino acid starvation with EBSS, both ULK1 and LC3 puncta formed in the presence of inhibitor, although their number was slightly reduced compared with DMSO-treated controls (Fig 4A and B). In the presence of either of the ULK1 inhibitors, the autophagosomal structures appeared to be brighter in intensity and indeed their size was significantly larger than those formed under control conditions (Fig 4A and B). In addition, both ULK1 and LC3 were decorating most of the enlarged autophagosomes (Fig 4A and B), implying stalled early autophagosomes/phagophores. These data were also confirmed by co-staining with WIPI2 and GABARAPL1, another early and early/late autophagosomal marker, respectively (Fig 4C and D). Large LC3 puncta were still apparent after 8 h of EBSS starvation with inhibitors, although further work is needed to confirm the actual half-life of these structures (Fig S5). Thus, the observations initially made with MRT68921 (Figs 1–3 and (24)) appear common to all three distinct ULK1 inhibitors, strongly implying that in addition to its upstream signalling roles, ULK1 catalytic activity is required for autophagosome maturation.

### VPS34 activation can occur following ULK1 inhibition

A key ULK1 substrate is the VPS34 complex (13, 14, 15, 25). However, the presence of stalled autophagosomal structures upon ULK1 inhibition, which are positive for the PI3P-binding proteins DFCP1 (Fig 2) or WIPI2 (Fig 4), suggests that VPS34 is still able to be activated despite ULK1 inhibition. To confirm this, we examined whether VPS34 inhibition could abolish WIPI2 recruitment to the ULK1-inhibited structures. We treated cells with EBSS or EBSS in combination with either MRT68921 or ULK-101 in the presence or absence of one of the most selective and potent VPS34 inhibitor available to date, VPS34-IN1 (31). As shown in Fig 5, the presence of VPS34-IN1 abolished WIPI2 puncta formation in EBSS-only treated cells as well as those treated in combination with either MRT68921 or ULK-101 (Fig 5A and B). These data suggest that upon ULK1 inhibition, VPS34 activity (and concomitant PI3P production) is still present and is essential to recruit WIPI2 to autophagosome forming sites. Thus, there are additional factors, independent of ULK1 catalytic activity, that are required for VPS34 activity at the phagophore. In the absence of ULK1 inhibition, VPS34-IN1 also prevented EBSS-induced ULK1 puncta accumulation, implying a potential feedback mechanism, although we cannot rule out that puncta still form but are smaller and harder to distinguish against the high background staining of the primary antibody used to detect ULK1 (Fig 5A and B). In support of the latter, enlarged ULK1 puncta were still forming in cells treated with both VPS34-IN1 and MRT68921 or ULK-101 (although with ULK-101 there appeared fewer in total), strongly suggesting that ULK1 is upstream of VPS34 (Fig 5A and B and S6). Similarly, and in support of the requirement for VPS34 activity in LC3 lipidation, stalled LC3-positive autophagosomes

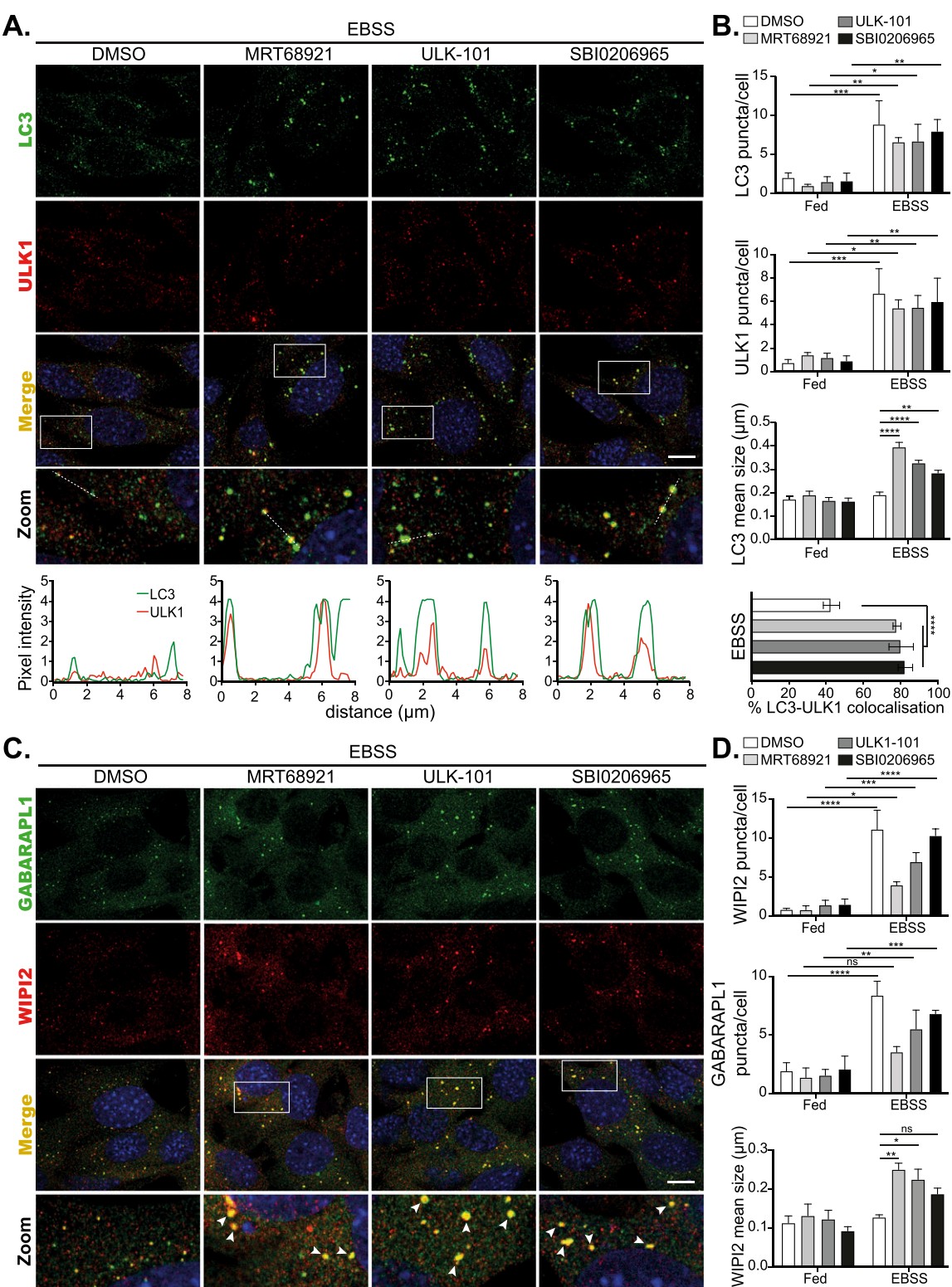

**Figure 4. Distinct ULK1 inhibitors lead to aberrant autophagosomal structures.**
**(A)** MEF cells were pretreated for 15 min with either 2 µM MRT68921, or 1 µM ULK-101, or 5 µM SBI0206965 followed by treatment with EBSS for 1 h in the presence or absence of the inhibitors at the above concentrations. Cells were fixed and stained for endogenous ULK1 (red), LC3 (green), and DAPI (blue) and imaged with a ZEISS LSM 880 confocal microscope. Scale bar, 10 µm. Boxed area is shown enlarged in zoom panels and dotted line is shown as a line scan below to highlight colocalization and puncta intensity. **(B)** Quantitation of indicated data from (A) shown as mean of n = 3 ± SEM. Statistical analysis was performed with a two-way ANOVA and a Tukey's multiple comparisons test (or a two-way ANOVA and a Sidak's multiple comparisons test for puncta size) Note, method of quantitation of LC3 mean size was distinct from

induced in the presence of MRT68921 were abolished in cells treated with VPS34-IN1 in combination with MRT68921 (Fig S6). Thus, although ULK1 may activate the VPS34 complex upon autophagy stimulation, this is not an absolute requirement for PI3P production and LC3 lipidation during autophagy stimulation. Taken together, our data show that not only is ULK1 catalytic activity essential for starvation-induced autophagy, but that it also plays multiple roles at distinct stages during the autophagosome maturation. It is not simply involved in VPS34 activation, but also functions downstream of this to regulate autophagosome formation and maturation, prior to release from the omegasome.

## Discussion

Specific autophagy inhibitors have both clinical potential for the treatment of autophagy-dependent cancers but can also act as key tools to study the function of ULK1 and autophagy in different contexts (32). So far, the most commonly used autophagy inhibitors are Bafilomycin A1 and Chloroquine (or its derivative hydroxychloroquine), which inhibit lysosomal activity and concomitantly autophagy (33, 34, 35, 36). As a result, these compounds have major effects on the endo-lysosomal pathway and thus do not allow discrimination between the different autophagy pathways (microautophagy, macroautophagy and chaperone-mediated autophagy) or lysosomal dysfunction per se. Given that ULK1/2 are essential for autophagy initiation at both the cell culture and organismal level (5), ULK1 is a promising target for the development of autophagy inhibitors (37). Being the most upstream kinase regulating autophagy, one may expect abolishment of autophagosome structures upon treatment with ULK1 inhibitors. However, we show that pharmacological inhibition of ULK1, with multiple distinct inhibitors, results in aberrant accumulation of enlarged autophagosomal structures decorated with early markers such as DFCP1 and WIPI2, as well as the later markers LC3 and GABARAPL1. Importantly, a kinase-dead mutant of ULK1 partially recapitulated this phenotype, further corroborating these findings. ULK1 can phosphorylate components of the VPS34 complex, including VPS34 and Beclin-1 and these phosphorylation events are thought to enhance VPS34 activity to promote autophagosome biogenesis (13, 14, 15, 25). Our data show that VPS34 is still able to drive PI3P formation upon ULK1 inhibition, as shown by recruitment of the PI3P-binding proteins DFCP1 and WIPI2 to sites of forming autophagosomes. Therefore, ULK1 may fine-tune VPS34 activity during autophagosome formation, but our data suggests these ULK1-mediated phosphorylation events are not an absolute requirement for VPS34 activation during autophagy stimulation. Indeed, seemingly ULK1-independent VPS34 activation mechanisms have been proposed (38).

Interestingly, loss of ULK1 (and its homologue ULK2) does not result in formation of stalled autophagosomes, but rather a block in their formation (see Fig 2 and (39, 40)). These important observations show that the ULK1 complex has both kinase-dependent and independent functions in autophagy. This is similar to the scenario observed in yeast, where a kinase-dead mutant of Atg1 (the yeast

homologue of ULK1) resulted in accumulation of autophagy markers at the pre-autophagosomal structure (PAS) and failure of their dissociation, which is required for autophagy progression (41, 44). In the same study, it was highlighted that Atg1 initiates autophagy by recruiting downstream proteins to the PAS in a kinase-independent manner; however, its role in the completion of later stages does depend on its kinase activity (41, 44). These data are similar to our observations, where inhibition of ULK1 kinase activity does not block autophagosome formation but results in accumulation of aberrant early structures. This suggests that the kinase activity–related functions of Atg1 and ULK1 are highly conserved and we speculate that a key role is to drive autophagosome release from the PAS/phagophore by phosphorylating, as yet unknown, substrate(s). In further support for multiple roles, ULK1 has been recently shown to regulate autophagosome-lysosome fusion by recruiting STX17 in a PKCα-dependent mechanism (42).

We recently published a study involving two novel ULK1 inhibitor scaffolds that abolished autophagosome formation (43). However, these compounds display promiscuous selectivity profiles in contrast to the inhibitors used here, implying additional kinases may regulate autophagosome formation (43). This highlights the importance of selectivity in understanding the mechanisms by which a kinase regulates a pathway and in avoiding off-target effects that might lead to artefacts. Our use of three ULK1 inhibitors in this study, with distinct selectivity profiles, circumvents this problem. Overall, our findings show that when ULK1 kinase activity is inhibited in cells, autophagosome formation is still primed but fails in later stages, resulting in aberrant accumulation of stalled autophagosomes. The key substrate(s), the phosphorylation of which is essential for avoiding this phenotype, are yet to be identified.

## Materials and Methods

### Chemicals, antibodies and DNA constructs

MRT68921 and VPS34-IN1 were obtained from Medical Research Council (MRC) Protein Phosphorylation and Ubiquitylation Unit (PPU) Reagents and Services. ULK-101 was purchased from MedChemExpress EU. SBI0206965 and Polybrene were from Sigma-Aldrich. Bafilomycin A1 was purchased from Enzo Life Sciences. The following antibodies were purchased from CST: ULK1 (8054S), pS757 ULK1 (6888T), pS555 ULK1 (5869S) and LC3 A/B (4108S—used for Western blot). pS318-ATG13 (NBP2-19127) was from Novus Biologicals. ATG13 (SAB4200100) was from Sigma-Aldrich. GABARAPL1 (ab86497) and Vinculin (ab129002) were from Abcam. FIP200 (17250-1-AP), α-Tubulin (66031-1-IG), and GAPDH (10494-1-AP) were purchased from Proteintech. The WIPI2 antibody (MCA5780GA) was purchased from Bio-Rad. The LC3 antibody (M152-3) used for endogenous immunofluorescence staining was purchased from MBL. The mCherry antibody (6g6-100) was from ChromoTek. The ULK1 antibody used for immunoprecipitation (DU 34016) and the DNA constructs used in this study (Flag-ULK1 [DU45617], Flag-ULK1 K46I [DU45618], mCherry-DFCP1 [DU24974],

---

that used in Fig 2 (see the Materials and Methods section for more detail). **(C, D)** as in (A, B) but cells were stained for endogenous WIPI2 (green) and GABARAPL1 (red). Arrowheads mark examples of abnormal WIPI2-GABARAPL1 autophagic structures. * = $P < 0.05$, ** = $P < 0.01$, *** = $P < 0.001$, **** = $P < 0.0001$ and ns, nonsignificant.

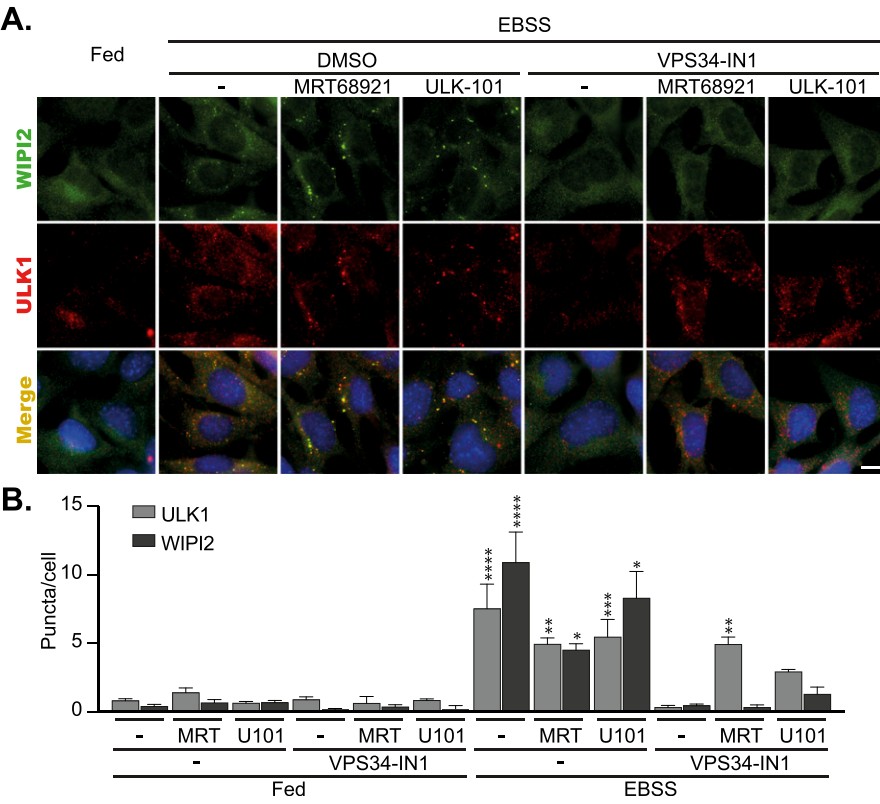

**A.**

**B.**

**Figure 5. VPS34 is still active after ULK1 inhibition.**
**(A)** MEF cells were pre-treated for 15 min with either 2 $\mu$M MRT68921, or 1 $\mu$M VPS34-IN1, or 1 $\mu$M ULK-101 followed by treatment with EBSS for 1 h in the presence or absence of the inhibitors at the above concentrations. After fixation, the cells were stained for endogenous ULK1 (red), WIPI2 (green) and DAPI (blue) and imaged using a Nikon Eclipse Ti wide-field microscope. Scale bar, 10 $\mu$m. **(B)** Quantitation of data shown in A representing mean of n = 3 ± SEM. Statistical analysis was performed with a two-way ANOVA and a Sidak's multiple comparisons test. * = $P < 0.05$, ** = $P < 0.01$, *** = $P < 0.001$, **** = $P < 0.0001$ and ns, nonsignificant. Comparisons of each treatment are with the relevant DMSO control.

GFP-LC3b [DU40253] and mCherry-GFP-LC3 [DU55696]) were generated at the MRC PPU Reagents and Services and are available to purchase online https://mrcppureagents.dundee.ac.uk.

### Cell culture and treatments

Immortalised MEF cells *ULK1/2* WT or DKO were a gift from Dr Craig Thompson (Memorial Sloan-Kettering Cancer Center) (39). Cells were cultured in DMEM supplemented with 10% (vol/vol) FBS (Sigma-Aldrich), 2 mM L-glutamine, 100 U/ml penicillin, and 0.1 mg/ml streptomycin (Thermo Fisher Scientific) (fed conditions), in a humidified incubator, with 5% $CO_2$ at 37° C. For treatments cells were at ~90% confluency for Western blotting or 60–70% confluency for imaging on the day of the experiment. For treatments, cells were pretreated with DMSO or each inhibitor for 15 min (except for Fig 1E, where pre-treatment was for 60 min) followed by further incubation or starvation for the time points indicated in figure legends. For starvation cells were washed twice in EBSS, followed by incubation in EBSS-only or EBSS and 50 nM BafA1 in the presence or absence of the inhibitors where indicated. The concentrations used for each compound in the experiments are indicated in figure legends.

### Stable cell line generation

To generate the retroviruses carrying the gene of interest, DNA constructs in pBABE vectors along with constructs for VSV-G (vesicular stomatitis virus G protein) and Gag-Pol (both from Takara Bio) were co-transfected in HEK293-FT cells. 6 $\mu$g of the pBABE plasmid, 3.8 $\mu$g of Gag-

Pol, and 2.2 $\mu$g of VSV-G were combined with 36 $\mu$l Lipofectamine 2000 in 600 $\mu$l Opti-MEM (both from Thermo Fisher Scientific) for 20 min in RT. For the transfection, the complexes were applied onto a 70% confluent dish of HEK293-FT cells along with 5 ml of Opti-MEM medium. Opti-MEM medium was replaced with fresh DMEM culture medium 5 h later. The next morning, the medium was again replaced with 10 ml fresh culture medium. 24 h later, the medium containing formed viral particles was collected and filtered through a 0.45 $\mu$m pore size filter and applied to MEF cells along with 10 $\mu$g/ml polybrene to increase the efficiency of infection. After 24 h, the medium was replaced with fresh medium and the next day cells were selected either with culture medium containing 2 $\mu$g/ml puromycin or with 100 $\mu$g/ml hygromycin.

### Live imaging

For live cell imaging, MEF *ULK1/2* DKO cells expressing Flag-ULK1, mCherry-DFCP1, and GFP-LC3 were plated onto a four compartment 35-mm glass bottom dish (Ibidi) 24 h before imaging. On the day of imaging, the cells were washed twice with pre-warmed EBSS followed by treatment with EBSS or EBSS and 2 $\mu$M MRT68921 and were immediately transferred to the microscope stage. Two to three fields were picked and imaged every 1 min for a total of 60 min using a Nikon Eclipse Ti wide-field microscope.

### Immunofluorescence staining

For immunofluorescence experiments cells were grown on 16-mm round glass coverslips for 24 h before treatments. Upon treatments,

the cells were washed in PBS followed by fixation with 3.7% PFA/10 mM Hepes pH 7.0. PFA was quenched in DMEM/10 mM Hepes, pH 7.0/0.02% $NaN_3$ for 20 min in RT. Cells were next permeabilised in 0.2% NP-40/PBS for 10 min and subjected to blocking with 1% BSA/PBS/0.01% $NaN_3$ (blocking buffer) for 30 min. Primary antibodies were diluted in blocking buffer (1:200 for ULK1 and LC3, 1:500 for WIPI2 and GABARAPL1), applied onto coverslips and incubated at 37°C in a humidified chamber for 60 min. Coverslips were then washed three times (5 min each) with blocking buffer. Secondary antibodies were applied in 1:500 dilution (in blocking buffer) and incubated for 30 min at RT, followed by three washes in blocking buffer (5 min each). Coverslips were then mounted onto glass slides with ProLong Gold Antifade Mountant with DAPI (Thermo Fisher Scientific). Images were taken using a wide-field Nikon Eclipse Ti wide-field microscope (Figs 1, 2, 5, S5, and S6) or a ZEISS 880 confocal scanning microscope (rest of figures).

### Western blotting

For Western blotting cells were washed twice in ice-cold PBS and were subsequently scraped in lysis buffer (50 mM Hepes, pH 7.4, 150 mM NaCl, 1 mM EDTA, 10% Glycerol, 0.5% NP-40, and protease/phosphatase inhibitor cocktails as described previously (43)). Lysates were incubated on ice for 20 min, following centrifugation at 20,000*g* at 4°C. The supernatant was then transferred to a new tube and lysates were subjected to protein content estimation using Bio-Rad Protein Assay Dye Reagent Concentrate (Bio-Rad), according to the manufacturer's instructions. Samples were prepared in 1× (LDS) lithium dodecyl sulfate buffer (Thermo Fisher Scientific) before loading on homemade 8% (or 13% for LC3 blot in Fig 1E) Tris-glycine gels or 10% Bis-Tris gels (for LC3 blot in Fig 3A only) for electrophoresis following standard protocols. Gels were next subjected to wet transfer onto PVDF membranes. Membranes were blocked in 5% milk in TBS-Tween for 30 min in RT, following washes in TBS-Tween and primary antibody incubation O/N at 4°C. Antibody dilutions were as follows: for ULK1, pS757 ULK1, pS555 ULK1, and FIP200 1:1,000 in 5% BSA/TBS-Tween, for pS318-ATG13 and LC3 1:1,000 in 5% milk/TBS-Tween, for ATG13 1:5,000 in 5% milk/TBS-Tween, for mCherry 1:2,000 in 5% BSA/TBS-Tween, for GAPDH 1:10,000 in 5% BSA/TBS-Tween, for Tubulin 1:40,000 in 5% BSA/TBS-Tween. Upon primary antibody incubation, membranes were washed three times in TBS-Tween, followed by secondary antibody (Thermo Fisher Scientific) incubation for 60 min in RT. Membranes were then washed three times in TBS-Tween, followed by development on hypersensitive X-ray films (Fig 1E only) or with a Chemidoc imaging system (Bio-Rad).

### Immunoprecipitation

For co-immunoprecipitation the cells were lysed in lysis buffer and 1 mg of total protein was incubated with 7 μg ULK1 antibody for 60 min at 4°C under rotation. 50 μl slurry of Protein A/G Resin (Expedeon) were added, followed by incubation for 60 min at 4°C under rotation. Beads were then washed three times in PBS and once in lysis buffer and proteins were eluted from beads with 2× LDS, followed by heating at 95°C for 5 min. Eluates were passed through a Spin-X column (Sigma-Aldrich) before loading on an 8% Tris-glycine gel and Western blot analysis.

### Flow cytometry

WT MEF cells, stably expressing mcherry-GFP-LC3, were grown on 6 cm dishes until reaching 70% confluency and treated with either DMSO or ULK1 inhibitors for 4 h with MRT68921 (4 μM), or ULK1-101 (2 μM), or SBI0206965 (20 μM) or Bafilomycin A (50 nM) in presence or absence of EBSS. Cells were harvested for analysis by washing once with PBS, followed by trypsinisation with Trypsin–EDTA (0.25%) (Thermo Fisher Scientific). The cells were then fixed in 3.7% PFA/10 mM Hepes, pH 7.0, for 15 min and finally re-suspended in 0.4 ml of Dulbecco's PBS containing 1% FBS into 5 ml Falcon round-bottom polystyrene test tubes 12 × 75 mm (Thermo Fisher Scientific).

Flow cytometry data were acquired on an LSR Fortessa II with DIVA software (BD Biosciences). Cells were gated according to their forward- and side-scatter profiles. 488-nm laser was used to detect GFP in emission filter 530/30 and 561 nm laser to detect mCherry in emission filter 610/20. Data were analysed using FlowJo software v10.7.1 (BD Biosciences). 20–50,000 cells were analysed per condition, with fluorescent detection in green and red channels. Increased autophagy was determined for individual cells by detecting decreased green versus red fluorescence, based on gating determined by the green and red fluorescence of vehicle (DMSO)–treated control cells.

### Quantification and statistical analysis

For quantification, a minimum of three independent experiments were included in the representing graphs as indicated in figure legends. For Western blot quantification, band densitometry analysis was performed in ImageJ. Phosphorylated levels of proteins were normalised to the total levels and LC3-II levels were normalised to loading control, as indicated on the graphs. In Fig 2B, autophagosome puncta number was quantified manually and autophagosome size was measured using the "area" tool in NIS Elements Imaging Software 4.20 (Nikon) and in Fig 4B and D using Volocity 6.3. Statistical analysis was performed with one- or two-way ANOVA in GraphPad Prism 8, and an appropriate post-test, as indicated in figure legends. Line scans were performed by plotting the intensity profiles of the lines in ImageJ.

# Supplementary Information

# Acknowledgements

We would like to thank the Medical Research Council Protein Phosphorylation and Ubiquitylation Unit Reagents and Services for DNA cloning and the Ganley laboratory members for reading and providing critical input for this manuscript. This work was funded by a grant from the Medical Research Council, UK (MC_UU_00018/2).

## Author Contributions

M Zachari: conceptualization, formal analysis, validation, investigation, visualization, methodology, and writing—original draft.
M Longo: formal analysis, validation, investigation, visualization, methodology, and writing—review and editing.
IG Ganley: conceptualization, formal analysis, supervision, funding acquisition, investigation, project administration, and writing—original draft, review, and editing.

## Conflict of Interest Statement

The authors declare that they have no conflict of interest.

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
