## [Reviewer comments · Life Science Alliance]

Life Science Alliance

Aberrant Autophagosome Formation Occurs Upon Small Molecule Inhibition Of ULK1 Kinase Activity

Maria Zachari, Marianna Longo, and Ian Ganley
DOI: <https://doi.org/10.26508/lsa.202000815>

Corresponding author(s): Ian Ganley, University of Dundee

Review Timeline:	Submission Date:	2020-06-16
	Editorial Decision:	2020-07-09
	Revision Received:	2020-10-06
	Editorial Decision:	2020-10-12
	Revision Received:	2020-10-19
	Accepted:	2020-10-19

Scientific Editor: Shachi Bhatt

Transaction Report:

July 9, 2020

Re: Life Science Alliance manuscript #LSA-2020-00815

Dr. Ian G. Ganley
University of Dundee
MRC Protein Phosphorylation and Ubiquitylation Unit
Sir James Black Centre, School of Life Sciences
Dow Street
Dundee, Scotland DD1 5EH
United Kingdom

Dear Dr. Ganley,

Thank you for submitting your manuscript entitled "Aberrant Autophagosome Formation Occurs Upon Small Molecule Inhibition Of ULK1 Kinase Activity" to Life Science Alliance. The manuscript was assessed by expert reviewers, whose comments are appended to this letter.

As you will see, the reviewers appreciate the significance of the findings presented in your manuscript. Although they also raise some points that need to be addressed, given the overall high level of interest in your study we would like to invite you to submit a revised version. When submitting the revision, please include a letter addressing the reviewers' comments point by point. Although we feel that a majority of the points that have been raised can be addressed, we would be happy to discuss individual revision points further with you should this be helpful. Please note that papers are generally considered through only one revision cycle, so strong support from the referees on the revised version is needed for acceptance.

In our view these revisions should typically be achievable in around 3 months. However, we are aware that many laboratories cannot function fully during the current COVID-19/SARS-CoV-2 pandemic and therefore encourage you to take the time necessary to revise the manuscript to the extent requested above. We will extend our 'scooping protection policy' to the full revision period required. If you do see another paper with related content published elsewhere, nonetheless contact me immediately so that we can discuss the best way to proceed.

Thank you for this interesting contribution to Life Science Alliance. We are looking forward to receiving your revised manuscript.

Sincerely,

Reilly Lorenz
Editorial Office Life Science Alliance
Meyerhofstr. 1
69117 Heidelberg, Germany
t +49 6221 8891 414
e contact@life-science-alliance.org
www.life-science-alliance.org

B. MANUSCRIPT ORGANIZATION AND FORMATTING:

Reviewer #1 (Comments to the Authors (Required)):

In the manuscript by Zacharia and Ganley the authors study the role of ULK1 in starvation induced autophagy using small molecule inhibitors of ULK1 kinase activity. They find that inhibition of ULK1

impairs the process of autophagosome formation not only at the nucleation stage but also at later stages. They further find that ULK1 activity is not strictly required for Vps34 activation, PI3P production and WIPI puncta formation. The paper presents interesting insights for scientists working in the field of autophagy and in particular those studying the process of autophagosome formation. The data mostly support the authors' conclusion, although a more quantitative approach would help to better support the author statements. In addition, as detailed below some of the statements related to the inhibition of autophagic flux and the stage(s) at which autophagosome formation is blocked upon inhibitor treatment should be toned down or backed up with additional experiments.

Major points:

- 1) In relation to figure 1E the authors state that inhibition of ULK1 blocks autophagic flux (line 212). According to the gel in figure 1E, LC3B-II is still detectable upon ULK1 inhibition, likely suggesting a slower flux, but not a complete blockage. Quantification of LC3B-II levels, perhaps in combination with analysis of p62 levels, would allow a better assessment of autophagy flux.
- 2) Figure 2: the authors state that in ULK1/2 DKO cells LC3 puncta are vastly diminished (lines 231-233). Quantification of the number of LC3 puncta/cell would better support this statement. Moreover, it is not clear what is the phenotype of the DKO cell lines, in absence of ULK1 inhibitor treatment. This panel should be added to the figure. In addition, the size of the autophagosomes could be measured and quantified.
- 3) In lines 246-247 it is stated that, upon ULK1 inhibition, the observed autophagosome-like structures do not traffic to the lysosome. This statement is not supported by data. The accumulation of bigger autophagosome-like structures could be due to a slower process of autophagosome formation which leads to the accumulation of more autophagy related proteins, but these structures might ultimately still traffic to lysosomes.
- 4) In line 269 the authors say that treatment with ULK1 inhibitors inhibits autophagic flux. According to the quantification of LC3-II levels, LC3B accumulation is still visible upon Bafilomycin treatment, although to a lesser extent than in WT cells. What can be concluded from the data is that the flux might be slower or decreased upon inhibition of ULK1, but not inhibited or blocked.

Minor points:

- 1) The authors should show a characterization of the stable cell lines used in Figure 1. For example, the expression levels of GFP-LC3 and mCherry-DFCP1, in comparison to WT proteins should be shown.
- 2) In line 202, when the authors describe the motility of LC3-positive structures, they should refer to the movie rather than the figure.
- 3) In line 213-214 (related to figure S1) the authors state that autophagosomes appearing upon ULK1 inhibition are positive for ATG2B. To support such a statement colocalization of ATG2B with LC3 or other autophagosomal marker should be shown. Alternatively, the supplementary figure could be removed, since it doesn't contribute to the message of the manuscript.
- 4) In the paragraph related to Figure 3 the effect on autophagy of different ULK1 inhibitors is compared. Although the effect of the inhibitors is well described towards the end of the section, the statements at the beginning of it (lines 263-265) are not very accurate. For example, the activity of AMPK, is actually affected by treatment with SBI0206965.
- 5) In lines 287-289 the autophagosomal structures obtained upon ULK1 inhibition are characterized in number and size. Their brightness is evaluated by tracing a line across 2 or more puncta in the figure. This quantification of the brightness does not reflect the whole population of puncta. A

better way of quantifying this parameter would be to measure the average signal intensities of ULK1 and LC3 puncta. Later in line 290 they say that ULK1 and LC3 decorates most of the enlarges autophagosomes. To better support this statement colocalization analysis of ULK1 and LC3 is needed.

6) Line 308 refers to Figure 5C which does not exist.

7) In figure S6 it is curious that upon starvation, treatment with Vps34 inhibitors lead to loss of ULK1 puncta, which then are restored upon treatment with ULK1 inhibitors. Maybe the authors could comment on these data.

Reviewer #2 (Comments to the Authors (Required)):

Summary

In this manuscript, Zachari and Ganley examined the induction phase of starvation-induced autophagy in mammalian cells. In particular, the authors sought to consolidate the role of ULK1's kinase activity in the formation of autophagosomes and their precursors using live-cell imaging and confocal microscopy. Remarkably, the authors found that treatment with three distinct ULK1 inhibitors only delayed the appearance of omegasomes and autophagosomes but did not abolished their formation. Consistent with previous findings, the authors showed that under these treatment conditions both otherwise transient autophagic structures persisted. Intriguingly, the authors found that these apparent stalled or incomplete autophagosomes were only present upon pharmacological or genetic inhibition but not upon complete loss of ULK1. Lastly, the authors went on to show that ULK1- and WIPI2-positive pre-autophagic structures formed dependent on the activity of the lipid kinase hVps34 but were independent of UKL1's kinase activity. Together, this work provides several important mechanistic insights on catalytic and non-catalytic functions of ULK1 with potential far reaching implications on our understanding of the signaling events during autophagosome biogenesis. While this is indeed an elegant, well-controlled and -rationalized study, a few points still need to be addressed:

1) Do these inhibitors disrupt the ULK1 complex?

2) Would reconstitution of ULK1/2 KO cells with an ULK1 variant that lack the kinase domain phenocopy the expression of catalytic inactive ULK1 in this setting?

3) The authors may want to consider a scenario in which ULK1's activity is required to prime a second kinase which then allows autophagosome formation to proceed. One of the known ULK1 targets might function in this regard. An obvious (and easy) candidate to test would be TBK1.

Reviewer #3 (Comments to the Authors (Required)):

Zachari et al present interesting findings analysing the effects of a ULK1 inhibitor, MRT68921, on autophagosome initiation and maturation. The data in this manuscript are robust, well quantified and are supported by previous findings from the same lab (Petherick et al, JBC, 2015). The authors extend their analyses to include multiple ULK1 inhibitors and compare those to the effects of inhibiting Vps34 activity on autophagosome biogenesis. They show that the ULK1 inhibitors (and kinase dead ULK1 mutant) do not disrupt LC3 lipidation and phagophore localization of various ATG proteins (including ULK1, WIPI2, ATG2, and LC3). However, later stages of autophagosome biogenesis appear to be disrupted by ULK1 kinase inhibition. Analysing the stages of autophagy disrupted by ULK1 inhibitors, in comparison to the frequently used Vps34 inhibitors, provide important tools to dissect autophagosome biogenesis and distinguish the specific relevance of ULK1 kinase activity.

I only have few minor comments that mainly involve textual changes and data analyses:

Figure S5: The conclusion on page 10 that the structures forming in the presence of ULK1 inhibitors are stable and still present after 8hrs of starvation is not well supported by this figure and in the absence of control images. Can the authors reanalyse the live imaging data in Figure 1 to measure the lifetime of LC3 puncta in the presence/absence of ULK1 inhibitors? Alternatively, the authors may simply alter the conclusions of this figure.

Figure 3A: Comparing the ratio of LC3-II between starved cells and cells starved in the presence of BafA1 to confirm that lysosome fusion is affected in the presence of the ULK1 inhibitor would strengthen this conclusion. This is a very interesting finding and could be further supported by LC3-LAMP1 colocalisation experiments or cargo degradation (e.g. p62), if possible.

The authors could use arrow heads to mark 1-2 puncta in the different channels of the live imaging movies (supplementary data) to ease their monitoring.

We appreciate all the Reviewers' work in helping to review our manuscript and we hope to have addressed their comments sufficiently in the revised manuscript version.

Reviewer #1 (Comments to the Authors (Required)):

In the manuscript by Zacharia and Ganley the authors study the role of ULK1 in starvation induced autophagy using small molecule inhibitors of ULK1 kinase activity. They find that inhibition of ULK1 impairs the process of autophagosome formation not only at the nucleation stage but also at later stages. They further find that ULK1 activity is not strictly required for Vps34 activation, PI3P production and WIPI puncta formation. The paper presents interesting insights for scientists working in the field of autophagy and in particular those studying the process of autophagosome formation. The data mostly support the authors' conclusion, although a more quantitative approach would help to better support the author statements. In addition, as detailed below some of the statements related to the inhibition of autophagic flux and the stage(s) at which autophagosome formation is blocked upon inhibitor treatment should be toned down or backed up with additional experiments.

- We would like to thank the Reviewer for their hard work in going through our manuscript and offering helpful comments. We believe that the manuscript is much more robust now.

Major points:

1) In relation to figure 1E the authors state that inhibition of ULK1 blocks autophagic flux (line 212). According to the gel in figure 1E, LC3B-II is still detectable upon ULK1 inhibition, likely suggesting a slower flux, but not a complete blockage. Quantification of LC3B-II levels, perhaps in combination with analysis of p62 levels, would allow a better assessment of autophagy flux.

- We agree with the Reviewer here and think in general there was perhaps a slight misunderstanding, likely due to a lack of precision in our terminology, which we apologise for. We assume the Reviewer is talking in absolute terms here and it is very rare that pharmacological inhibition results in a complete blockage of activity in cells, thus our terminology of inhibition and blockage was used a bit more loosely to incorporate a significant amount of inhibition while not necessarily meaning complete inhibition. We have changed the phrasing here slightly (and in other places) to hopefully clarify this:
"Inhibition of ULK1 with MRT68921 was confirmed by western blot analysis of phospho-ATG13 (at serine 318), a well characterised substrate of ULK1, and a significant block in LC3-II flux (Fig. 1E and F)."
- We have also quantified LC3 flux by western blot and included the data in the new Fig. 1F – as in our previous publication (Petherick et al., 2015) there is no significant increase in LC3 accumulation +/- Bafilomycin in the presence of MRT68921. We feel this data shows that LC3 flux is impaired. We also tried to analyse p62 flux, as

suggested by the reviewer, however with the MEFs used in this study, we found very little p62 flux under the time course used (regardless of whether inhibitor was present), implying that autophagy of p62 in these cells occurs at a slower rate compared to that of LC3. As another assay of autophagic flux, we carried out flow cytometry of tandem LC3 expressing cells (new Fig. 3C). As can be seen, all the inhibitors significantly impair autophagy.

- Fig1.

Fig.3

2) Figure 2: the authors state that in ULK1/2 DKO cells LC3 puncta are vastly diminished (lines 231-233). Quantification of the number of LC3 puncta/cell would better support this statement. Moreover, it is not clear what is the phenotype of the DKO cell lines, in absence of ULK1 inhibitor treatment. This panel should be added to the figure. In addition, the size of the autophagosomes could be measured and quantified.

- We apologise for this omission and have now included the control DKO panel and quantitation for LC3 puncta number and size – the numbers do indeed support our previous conclusions. We do note that the numbers here are slightly different from later quantitation (Fig. 4B); however, the cells here express exogenous ULK1 and quantitation was carried out on a different microscope (due to COVID restrictions we were not able to use the same microscope). This is made clear in the legend of Fig. 4 and Methods section.

- Fig.2

3) In lines 246-247 it is stated that, upon ULK1 inhibition, the observed autophagosome-like structures do not traffic to the lysosome. This statement is not supported by data. The accumulation of bigger autophagosome-like structures could be due to a slower process of autophagosome formation which leads to the accumulation of more autophagy related proteins, but these structures might ultimately still traffic to lysosomes.

- We apologise for the confusion here and have removed the statement that trafficking is blocked – we were referring to data with LC3 flux suggesting the structures are impaired in their lysosomal degradation (from this study and our previous one). Please also see response to point 1 above too in that we agree, the formation of autophagosomes is slowed, which in turn significantly impairs their flux. We have altered the text slightly to hopefully clarify this:
“In contrast, loss of ULK1 kinase activity (either by genetic modification or pharmacologically) still results in recruitment of the autophagic machinery and appearance of autophagosome-like structures, although these are aberrantly larger in size and display reduced lysosomal flux (Fig. 1E and F, Fig. 2, Fig 3 and [24]). This is in support of previously published data showing that catalytically dead ULK1 blocks autophagy [29, 30].”

4) In line 269 the authors say that treatment with ULK1 inhibitors inhibits autophagic flux. According to the quantification of LC3-II levels, LC3B accumulation is still visible upon Bafilomycin treatment, although to a lesser extent than in WT cells. What can be concluded from the data is that the flux might be slower or decreased upon inhibition of ULK1, but not inhibited or blocked.

- As with the above points, we have changed the phrasing slightly to make it clear we are not implying a complete block/inhibition. Text changed:
“Autophagic flux was also impaired by all of the three ULK1 inhibitors”

Minor points:

1) The authors should show a characterization of the stable cell lines used in Figure 1. For example, the expression levels of GFP-LC3 and mCherry-DFCP1, in comparison to WT proteins should be shown.

- We have now included a western blot of the cell line in a new Fig. S1. Unfortunately, the antibody we have for DFCP1 does not detect endogenous protein. We have included a blot for the exogenous mCherry-DFCP1, but as this is stable retroviral-mediated expression (usually low) and we are not comparing to non mCherry-DFCP1-expressing cells, we feel this does not impact our conclusions.
- Fig. S1

2) In line 202, when the authors describe the motility of LC3-positive structures, they should refer to the movie rather than the figure.

- We have changed the text to reflect this:
 “Following dissociation, the autophagosomes remained positive for GFP-LC3 and motile in the cytoplasm (Movie S1 and Fig. 1B-D).”

3) In line 213-214 (related to figure S1) the authors state that autophagosomes appearing upon ULK1 inhibition are positive for ATG2B. To support such a statement colocalization of ATG2B with LC3 or other autophagosomal marker should be shown. Alternatively, the supplementary figure could be removed, since it doesn’t contribute to the message of the manuscript.

- On the Reviewer’s advice, we have removed this figure.

4) In the paragraph related to Figure 3 the effect on autophagy of different ULK1 inhibitors is compared. Although the effect of the inhibitors is well described towards the end of the section, the statements at the beginning of it (lines 263-265) are not very accurate. For example, the activity of AMPK, is actually affected by treatment with SBI0206965.

- We apologise for the confusion and have now altered the text to read:
 “Initially, we performed immunoblotting to identify concentrations and timepoints where ULK1 activity is sufficiently suppressed and LC3 flux is inhibited, as well as to confirm the activity status of upstream autophagy-regulating kinases (mTORC1 and AMPK).”
- And:
 “A non-significant reduction in pS555 ULK1 levels was observed upon SBI0206965 treatment in Fed conditions, implying potential AMPK inhibition (Fig. 3A and S3B), which has also been reported elsewhere [32]. However, given that AMPK phosphorylation of ULK1 (pS555) is dramatically reduced under these autophagy-inducing conditions, we assume that any AMPK inhibitory effects of SBI0206965 will have a negligible impact on ULK1 in this instance.”

5) In lines 287-289 the autophagosomal structures obtained upon ULK1 inhibition are characterized in number and size. Their brightness is evaluated by tracing a line across 2 or more puncta in the figure. This quantification of the brightness does not reflect the whole population of puncta. A better way of quantifying this parameter would be to measure the average signal intensities of ULK1 and LC3 puncta. Later in line 290 they say that ULK1 and LC3 decorates most of the enlarges autophagosomes. To better support this statement colocalization analysis of ULK1 and LC3 is needed.

- We have now included quantitation of LC3 and ULK1 co-localisation across the whole population (new Fig. 4B). We have kept the line scans as we feel they offer a qualitative view of colocalization that is complementary to the new quantitative data.

- Fig.4

6) Line 308 refers to Figure 5C which does not exist.

- We are sorry for this mistake, it has been corrected to 5B.

7) In figure S6 it is curious that upon starvation, treatment with Vps34 inhibitors lead to loss of ULK1 puncta, which then are restored upon treatment with ULK1 inhibitors. Maybe the authors could comment on these data.

- We agree with the Reviewer that this is intriguing data, yet we cannot fully explain this at the moment. We have added the following to the text where the Figure is described:

“In the absence of ULK1 inhibition, VPS34-IN1 also prevented ULK1 puncta accumulation, though we cannot rule out that puncta still form but are smaller and harder to distinguish against the high background staining of the primary antibody used to detect ULK1 (Fig. 5A and B). In support of the latter, enlarged ULK1 puncta were still forming in cells treated with both VPS34-IN1 and MRT68921 or ULK-101 (although with ULK-101 there appeared fewer in total), strongly suggesting that ULK1 is upstream of VPS34 (Fig. 5A, B and S6).”

Reviewer #2 (Comments to the Authors (Required)):

Summary

In this manuscript, Zachari and Ganley examined the induction phase of starvation-induced autophagy in mammalian cells. In particular, the authors sought to consolidate the role of ULK1's kinase activity in the formation of

autophagosomes and their precursors using live-cell imaging and confocal microscopy. Remarkably, the authors found that treatment with three distinct ULK1 inhibitors only delayed the appearance of omegasomes and autophagosomes but did not abolish their formation. Consistent with previous findings, the authors showed that under these treatment conditions both otherwise transient autophagic structures persisted. Intriguingly, the authors found that these apparent stalled or incomplete autophagosomes were only present upon pharmacological or genetic inhibition but not upon complete loss of ULK1. Lastly, the authors went on to show that ULK1- and WIPI2-positive pre-autophagic structures formed dependent on the activity of the lipid kinase hVps34 but were independent of ULK1's kinase activity. Together, this work provides several important mechanistic insights on catalytic and non-catalytic functions of ULK1 with potential far reaching implications on our understanding of the signaling events during autophagosome biogenesis. While this is indeed an elegant, well-controlled and -rationalized study, a few points still need to be addressed:

- We thank the Reviewer for their supportive comments and thorough examination of our manuscript.

1) Do these inhibitors disrupt the ULK1 complex?

- We thank the Reviewer for raising this important point. We carried out an endogenous ULK1 co-IP from cells treated with inhibitors and found that complex formation was similar among all treatments. The new data is shown in Fig. S3C and mentioned in the text:

“In addition, we found that treatment of cells with the three inhibitors did not appreciably disrupt the ULK1 complex itself, as determined by co-IP of ATG13 and FIP200 with ULK1 (Fig. S3C).”

- Fig. S3

C.

2) Would reconstitution of ULK1/2 KO cells with an ULK1 variant that lack the kinase domain phenocopy the expression of catalytic inactive ULK1 in this setting?

- This was an interesting suggestion and we tried to express an ULK1 truncation mutant lacking the kinase domain. However, we found it to be very poorly expressed in preliminary work, thus we abandoned this study as it would make interpretation

of the results difficult due to inconsistencies in ULK1 protein expression between constructs.

3) The authors may want to consider a scenario in which ULK1's activity is required to prime a second kinase which then allows autophagosome formation to proceed. One of the known ULK1 targets might function in this regard. An obvious (and easy) candidate to test would be TBK1.

- We agree with the Reviewer on this point, which was actually a major part of the first author's PhD project. However, we have been unsuccessful to date in identifying the direct target of ULK1 in this instance. We had previously looked at TBK1 in our first study, but found that ULK1 inhibition still resulted in large puncta in double TBK1/IKKe knockout MEFs, implying TBK1 is not the target (Petherick et al., 2015).

Reviewer #3 (Comments to the Authors (Required)):

Zachari et al present interesting findings analysing the effects of a ULK1 inhibitor, MRT68921, on autophagosome initiation and maturation. The data in this manuscript are robust, well quantified and are supported by previous findings from the same lab (Petherick et al, JBC, 2015). The authors extend their analyses to include multiple ULK1 inhibitors and compare those to the effects of inhibiting Vps34 activity on autophagosome biogenesis. They show that the ULK1 inhibitors (and kinase dead ULK1 mutant) do not disrupt LC3 lipidation and phagophore localization of various ATG proteins (including ULK1, WIPI2, ATG2, and LC3). However, later stages of autophagosome biogenesis appear to be disrupted by ULK1 kinase inhibition. Analysing the stages of autophagy disrupted by ULK1 inhibitors, in comparison to the frequently used Vps34 inhibitors, provide important tools to dissect autophagosome biogenesis and distinguish the specific relevance of ULK1 kinase activity.

- We thank the Reviewer for all their time and effort in going through our manuscript.

I only have few minor comments that mainly involve textual changes and data analyses:

Figure S5: The conclusion on page 10 that the structures forming in the presence of ULK1 inhibitors are stable and still present after 8hrs of starvation is not well supported by this figure and in the absence of control images. Can the authors reanalyse the live imaging data in Figure 1 to measure the lifetime of LC3 puncta in the presence/absence of ULK1 inhibitors? Alternatively, the authors may simply alter the conclusions of this figure.

- Unfortunately we have found it tricky to image these structures over a long period of time – due to their movement, it requires taking frequent images (every 30 seconds)

to ensure that we image the same puncta. In our microscopy set-up, this results in phototoxicity over any period of time longer than an hour. In the timecourse of Fig. 1, the inhibited structures remained stable but we only imaged these structures for 30 mins on average.

We have altered the conclusions of this section to now read:

“Large LC3 puncta were still apparent following 8 hours of EBSS starvation with inhibitors, though further work is needed to confirm the actual half-life of these structures (Fig. S5).”

Figure 3A: Comparing the ratio of LC3-II between starved cells and cells starved in the presence of BafA1 to confirm that lysosome fusion is affected in the presence of the ULK1 inhibitor would strengthen this conclusion. This is a very interesting finding and could be further supported by LC3-LAMP1 colocalisation experiments or cargo degradation (e.g. p62), if possible.

- We thank the reviewer for this comment. Though we have not shown that fusion itself is blocked, we now have additional quantitative data that shows LC3-lysosomal flux is impaired upon ULK1 inhibition ((Fig. 1E-F as well as that in Fig. 3A and B). To further support this, we have tried p62 flux assays, but we have found that in MEFs, p62 turnover is slow compared to that of LC3, so in our experiments we have found very little change regardless of the presence or absence of inhibitor. To try and assess autophagic flux in another way, we expressed tandem mCherry-GFP-LC3 in our cells and monitored flux using flow cytometry (new Fig. 3C). As can be seen, autophagy is significantly impaired with all three inhibitors. Given that autophagosomal structures still form, and that with ULK1 inhibition there are more LC3 puncta that are positive for ULK1 (new quantitative data in Fig. 4B), we take this mean that there is a defect downstream of initiation but upstream of fusion. We do appreciate that further work is needed, beyond this current manuscript, to pinpoint the exact defect (and key ULK1 substrate identification) that occurs with ULK1 inhibition. We hope to address this in a future manuscript.

- Fig. 1

- Fig. 3

The authors could use arrow heads to mark 1-2 puncta in the different channels of the live imaging movies (supplementary data) to ease their monitoring.

- We have added arrows to the movies to highlight the structures shown in Fig. 1A and B.

October 12, 2020

RE: Life Science Alliance Manuscript #LSA-2020-00815R

Dr. Ian G. Ganley
University of Dundee
MRC Protein Phosphorylation and Ubiquitylation Unit
Sir James Black Centre, School of Life Sciences
Dow Street
Dundee, Scotland DD1 5EH
United Kingdom

Dear Dr. Ganley,

Thank you for submitting your revised manuscript entitled "Aberrant Autophagosome Formation Occurs Upon Small Molecule Inhibition Of ULK1 Kinase Activity". We would be happy to publish your paper in Life Science Alliance pending final revisions necessary to meet our formatting guidelines.

Along with the points listed below, please also attend to the following:

-please upload your supplementary figures as single files

A. FINAL FILES:

-- Summary blurb (enter in submission system): A short text summarizing in a single sentence the study (max. 200 characters including spaces). This text is used in conjunction with the titles of papers, hence should be informative and complementary to the title. It should describe the context and significance of the findings for a general readership; it should be written in the present tense

and refer to the work in the third person. Author names should not be mentioned.

B. MANUSCRIPT ORGANIZATION AND FORMATTING:

Sincerely,

Shachi Bhatt, Ph.D.
Executive Editor
Life Science Alliance
<https://www.life-science-alliance.org/>
Tweet @SciBhatt @LSAjournal

Reviewer #1 (Comments to the Authors (Required)):

The authors have addressed all my comments and I recommend publication.

Reviewer #2 (Comments to the Authors (Required)):

The authors adequately addressed all my concerns. Well done! Thus, I recommend to accept this manuscript for publication.

Reviewer #3 (Comments to the Authors (Required)):

The authors have addressed all my comments and the manuscript is suitable for publication.

October 19, 2020

RE: Life Science Alliance Manuscript #LSA-2020-00815RR

Dr. Ian G. Ganley
University of Dundee
MRC Protein Phosphorylation and Ubiquitylation Unit
Sir James Black Centre, School of Life Sciences
Dow Street
Dundee, Scotland DD1 5EH
United Kingdom

Dear Dr. Ganley,

Thank you for submitting your Research Article entitled "Aberrant Autophagosome Formation Occurs Upon Small Molecule Inhibition Of ULK1 Kinase Activity". It is a pleasure to let you know that your manuscript is now accepted for publication in Life Science Alliance. Congratulations on this interesting work.

*****IMPORTANT:** If you will be unreachable at any time, please provide us with the email address of an alternate author. Failure to respond to routine queries may lead to unavoidable delays in publication.*******

DISTRIBUTION OF MATERIALS:

Again, congratulations on a very nice paper. I hope you found the review process to be constructive and are pleased with how the manuscript was handled editorially. We look forward to future exciting

submissions from your lab.

Sincerely,

Shachi Bhatt, Ph.D.

Executive Editor

Life Science Alliance

<https://www.life-science-alliance.org/>
